# Cancer Therapy-Related Cardiovascular Complications in Clinical Practice: Current Perspectives

**DOI:** 10.3390/jcm10081647

**Published:** 2021-04-13

**Authors:** Michał Bohdan, Anna Kowalczys, Agnieszka Mickiewicz, Marcin Gruchała, Ewa Lewicka

**Affiliations:** 1First Department of Cardiology, Medical University of Gdańsk, 80-211 Gdańsk, Poland; anna.kowalczys@gumed.edu.pl (A.K.); agnieszka.mickiewicz@gumed.edu.pl (A.M.); marcin.gruchala@gumed.edu.pl (M.G.); 2Department of Cardiology and Electrotherapy, Medical University of Gdańsk, 80-211 Gdańsk, Poland; ewa.lewicka@gumed.edu.pl

**Keywords:** cardio-oncology, cardiotoxicity, heart failure, left ventricular dysfunction, comorbidities, cardiovascular disease, cancer, cancer treatment

## Abstract

Cardiovascular (CV) diseases and cancer are the leading causes of death in Europe and the United States. Both diseases have extensive overlap and share common risk factors, symptoms, and outcomes. As the number of patients with both cancer and CV diseases continues to rise, the field of cardio-oncology is gaining increased attention. A frequent problem during anti-cancer treatment is cardiotoxicity caused by the side-effects of chemo-, immuno-, targeted, and radiation therapies. This problem may manifest as acute coronary syndrome, myocarditis, arrhythmias, or heart failure. Modern cardio-oncology spans many different research areas. While some researchers focus on treating patients that have already developed cardiotoxicity, others aim to identify new methods for preventing cardiotoxicity before, during, and after anti-cancer therapy. Both groups share the common understanding that regular monitoring of cancer patients is the basis for optimal medical treatment. Optimal treatment can only be achieved through close cooperation between cardiologists and oncologists. This review summarizes the current views on cardio-oncology and discusses the cardiotoxicities associated with commonly used chemotherapeutics.

## 1. Introduction

Cancer is the second leading cause of death after cardiovascular (CV) diseases in the United States and Europe. Recent advances in anti-cancer treatments have led to some improvements in the management of patients with cancer, but mortality rates remain high [1]. Depending on the specific kind of tumor, up to 30% of patients die due to CV causes [2]. Among the growing number of cancer survivors, there is a notable increase in the prevalence of chemotherapy side-effects on the CV system [3]. CV diseases are among the main causes of long-term morbidity and mortality in cancer survivors [4,5,6]. Several types of anti-cancer therapies, namely, chemo-, immuno-, targeted, and radiation therapies, might cause cardiotoxicities. This review presents the current knowledge on the cardiotoxicities associated with all anti-cancer therapies, except radiotherapy.

Cardio-oncology aims to improve the management of cancer patients with therapy-related CV complications. Currently, the primary focuses of cardio-oncology include (1) preventive strategies in cancer patients with and without CV risk factors, (2) optimization of CV disease management, (3) early identification and treatment of CV cardiotoxicities, and (4) long-term CV follow-up for cancer survivors [7]. At present, cardio-oncology is considered one of the most rapidly expanding areas of medicine, not only for clinical studies, but also for basic research [4]. Many specialized cardio-oncology units have been established in Europe and the United States to assist the growing number of cancer patients at risk of developing CV complications or cancer therapy-related CV side-effects [7]. More research is needed to develop personalized strategies to improve CV outcomes in cancer patients and to enhance our understanding of the molecular crosstalk between cancer and the heart [8].

## 2. Common Chemotherapeutics and Their Mechanism of Cardiotoxicity

Several widely used anti-cancer drugs, such as anthracyclines, alkylating agents, fluoropyrimidines, human epidermal growth receptor type 2 (HER-2) antibodies, vascular endothelial growth factor (VEGF) inhibitors, and tyrosine kinase inhibitors (TKIs), may cause cardiotoxicity [4]. All these drugs may eventually lead to heart failure (HF) or other complications [4,9] (Figure 1). The molecular mechanisms leading to cardiotoxicity are diverse and often specific to each group of chemotherapeutics (Table 1).

### 2.1. Anthracyclines

Anthracyclines are widely used as part of anti-cancer regimens due to their beneficial effects on survival in patients with solid tumors and hematologic malignancies [19], particularly breast cancer, leukemias, sarcomas, lymphomas, and childhood tumors [20,21,22,23,24]. The cardiotoxic effects of anthracyclines are studied intensively in cardio-oncology, as the occurrence of HF or left ventricular (LV) dysfunction due to their administration has long been known. In fact, HF or LV dysfunction may occur in up to 48% of cancer patients who have been administered anthracyclines as part of their anti-cancer regimen [4]. The effects on the heart are dose-dependent: a total dose of 400 mg/m^2^ of doxorubicin is known to cause cardiotoxicity in 3–5% of cancer patients, whereas a total dose of 700 mg/m^2^ is known to cause cardiotoxicity in 18–48% of cancer patients. However, some authors suggest that there is no “safe” dose for doxorubicin and that a cumulative dose of 250 mg/m^2^ is potentially cardiotoxic [4]. Other risk factors include age (>65 or <18 years), female sex, kidney disease, other potential cardiotoxic chemo- or radiation therapies, and the presence of a CV disease [4].

The pathogenesis of anthracycline-induced cardiotoxicity is multifactorial. It includes free radical generation, iron metabolism, calcium homeostasis dysregulation, alterations in sarcomeric structure, and apoptosis. The literature shows that reactive oxygen species (ROS) induced by anthracyclines in particular have adverse effects on the heart. Additionally, topoisomerase II beta (TOP2β) is disabled, causing DNA double-strand breaks that ultimately lead to mitochondrial dysfunction, the activation of the p53 tumor suppressor protein, and the generation of ROS [25,26]. It has been shown that the genetic deletion of TOP2ß in mice leads to cardioprotective effects [27]. Moreover, the cardioprotective effects of dexrazoxane are mediated over the TOP2ß pathway [28]. Both pathways lead to cardiac cell death, which is responsible for the occurrence of cardiotoxicity.

Based on the time of onset, three different types of cardiotoxicities associated with anthracyclines have been distinguished: *acute* (occurring after a single dose), *early* (emerging within 1 year), and *late* (developing 1 or more years after the end of the treatment). The majority of patients treated with anthracyclines develop cardiac complications within one year after treatment termination [3,4].

### 2.2. Alkylating Agents

Cyclophosphamide is one of the important drugs used in the conditioning of patients undergoing hematopoietic stem cell transplantation [29]. In addition, alkylating agents are used in the treatment of patients with multiple myeloma and breast cancer and to treat malignancies in children [30,31]. The mechanism of cyclophosphamide action involves the alkylation of tumor DNA. The pathogenesis of cardiotoxicity associated with cyclophosphamide is not fully understood [32]. It is possible that ROS are involved in the process of cardiomyocyte apoptosis related to cyclophosphamide [23]. Moreover, cyclophosphamide cardiotoxicity is associated with inflammation, endothelial dysfunction, calcium alterations, endoplasmic reticulum, and mitochondrial damage [33]. It has been shown that cardiotoxicity (congestive HF, arrhythmias, cardiac tamponade, and myocardial dysfunction) occurs with cyclophosphamide treatment in 7–28% of cancer patients and is more frequently observed when high doses are administered [4,32]. HF related to cyclophosphamide is relatively rare and typically observed within days of drug administration [4].

### 2.3. Proteasome Inhibitors

Currently, proteasome inhibitors (PIs) play a major role in the treatment of multiple myeloma [4]. Unfortunately, inhibiting the function of the proteasomes responsible for the degradation of abnormal or dysfunctional proteins can disturb the protein homeostasis in cardiomyocytes, leading to their subsequent damage and apoptosis [18,34]. Cardiotoxicity has been thoroughly documented with regard to carfilzomib treatment and reported more frequently during the first 18 cycles of therapy than in later cycles [35]. The literature contains similar reports about bortezomib and ixazomib, which may indicate a class effect [18,35,36]. The most common CV complications observed after PI treatment are HF, hypertension, and arrhythmias [4,18,34]. HF affects 25% and 4% of patients treated with carfilzomib and bortezomib, respectively, and it most often occurs within the first weeks of chemotherapy [4]. Arterial hypertension is observed much less frequently in patients treated with PI: 6.5–12% of patients are affected [37]. An estimation of the initial CV risk, vigilance, and careful monitoring during treatment are important in the prevention of PI-related cardiotoxicity [38]. Patients at high risk of cardiotoxicity primarily include individuals >60 years of age, especially those previously treated with anthracyclines and those with multiple CV risk factors or amyloidosis [34].

### 2.4. Antimetabolites

Fluoropyrimidines (5-fluorouracil, capecitabine, gemcitabine) are mainly used in the treatment of solid tumors, particularly in colorectal, breast, pancreatic, and gastric cancer. The most common clinical manifestations of fluoropyrimidine-related cardiotoxicity include chest pain, angina, myocardial infarction, arrhythmias, and sudden cardiac death [39]. Chest pain occurs in approximately 1–18% of patients treated with 5-fluorouracil (5-FU) and in 0.5–8% of patients treated with its pro-drug capecitabine [40]. In rare cases, 5-FU can induce malignant ventricular tachycardia, cardiomyopathy, and heart failure with reduced ejection fraction (HFrEF) [41]. Symptoms of cardiotoxicity most often appear in the first few days after administration [11,39]. The main pathophysiological mechanism involves coronary vasospasm, thrombosis, and endothelial injury [39]. Other hypothesized mechanisms include direct myocardial injury, increased metabolism leading to energy depletion and ischemia, oxidative stress causing cellular damage, and a diminished ability of red blood cells to transfer oxygen, resulting in myocardial ischemia [11,39,41,42]. There is a higher risk of 5-FU cardiotoxicity in patients with pre-existing CV disease or renal impairment and in patients treated with continuous infusions and combined chemotherapy with alkylating agents [11,39,41].

### 2.5. Inhibitors of the HER-2 Signaling Pathway

One of the most frequently used inhibitors of the HER-2(/ErbB2) signaling pathway is the humanized monoclonal antibody trastuzumab. It is commonly used in HER-2–positive breast cancer and HER-2–positive gastric cancer [4]. The incidence of cardiotoxicity is estimated to be 2–20% and depends greatly on the concomitant treatment with other cardiotoxic anti-cancer drugs. In particular, anthracyclines significantly increase the risk of cardiotoxicity. Trastuzumab-related cardiotoxicity typically occurs during its administration [4]. The cardiotoxic effects are often reversible when trastuzumab treatment is stopped [4,43]. Banke et al. reported that trastuzumab treatment is also associated with a significant increase in the risk of late HF [44]. The mechanism of cardiotoxicity in the course of trastuzumab therapy is complex. It has been shown that the HER-2/ErbB2 receptor pathway activated by the growth factor neuregulin-1 plays an important protective role in reducing the sensitivity of cardiomyocytes to stress [22,25,45]. Notably, the HER-2/ErbB2 pathway also has an important effect on the pathogenesis of anthracycline-induced cardiotoxicity, which may explain the significantly higher rate of HF in patients treated with both anthracyclines and HER-2(/ErbB2) signaling pathway inhibitors [21,46]. Nevertheless, it should be highlighted that the highest rate of toxicity has been reported for combined, simultaneous therapy with anthracycline and HER-2 inhibition [4,47]. Slamon et al. observed that a non-anthracycline trastuzumab-based regimen in HER-2–positive breast cancer patients not only showed an efficacy similar to anthracycline concurrent with trastuzumab, but also resulted in fewer acute toxic effects, including lower risks of cardiotoxicity and leukemia [47].

### 2.6. Immune Checkpoint Inhibitor-Associated Cardiotoxicity

Immune checkpoint inhibitors (ICIs) are one of the most promising novel cancer therapy drugs. They have already been approved for use and are being studied in clinical trials focusing on different types of cancer (e.g., melanoma, non-small cell lung cancer, renal cell carcinoma, and Hodgkin’s lymphoma) [48,49,50,51]. This group of drugs inhibits T cell–cancer cell interactions and uncouples the “turn off” signals in these cells [52]. Therapy with these drugs results in several immune response phenomena [43]. The clinical cardiac phenotypes include arrhythmias, acute HF, hypotension in the presence of vasculitis, and peri- and myocarditis. Moreover, ICIs can promote rapid growth of atherosclerotic plaque through plaque-mediated T cell reactivation, infiltration and increased apoptosis of macrophages, and inhibition of the PD-1/PD-L1 pathways, thereby reducing the proatherogenic T cell response. These mechanisms lead to accelerated atherosclerosis and a 3-fold increase in the incidence of cardiovascular diseases, including stroke, coronary artery disease, and myocardial infarction requiring revascularization [53]. Myocarditis is considered the most severe complication of ICIs, and data indicates a median onset time of 18–39 days after the first dose [43,52,54,55]. The occurrence of myocarditis depends on the checkpoint inhibitor used and can affect up to 1.3% of patients treated with ICI combination therapy [44]. The rate of major adverse cardiac events in patients who develop myocarditis after immune treatment may reach 50% [56]. This specific type of myocarditis is associated with increased cardiac biomarkers; the roles of cardiac imaging, arrhythmia detection, troponin, and further cardiac evaluations (e.g., biopsies and stress tests) need to be determined in more detail. Patients with a diagnosis of ICI-induced myocarditis are typically first-line treated with steroids, with a recommended initial dose of 1–2 mg/kg of methylprednisolone [56,57,58]. Some authors found a significant association between high-dose steroid therapy and a reduction in major adverse CV event (MACE) occurrence [56,57]. Moreover, some studies support the implementation of other strategies, including immunosuppressive therapies, particularly in patients who do not respond to steroids (e.g., anti-thymocyte globulin, intravenous immunoglobulin, mycophenolate mofetil, and infliximab or rituximab) [43,44,59]. However, immunosuppressive treatment should be individualized for each patient treated for ICI-associated myocarditis. Notably, infliximab has been associated with HF and is contraindicated at high doses for this group of patients [59]. Moreover, some case reports and findings point to the usefulness of novel, specific immunosuppressive treatment for ICI-associated myocarditis. Esfahani et al. observed that alemtuzumab, a monoclonal antibody that binds to CD52, led to the rapid resolution of cardiac immune toxic effects in a 71-year-old woman being treated with first-line pembrolizumab for stage IV melanoma [60]. Salem et al. presented a clinical case of a 66-year-old woman with metastatic lung cancer, in whom the use of abatacept, a cytotoxic T-lymphocyte–associated antigen-4 agonist, led to the resolution of severe glucocorticoid-refractory myocarditis induced by ICI [61]. The key role of interleukin 6 in the pathogenesis of inflammation prompted some authors to use tocilizumab in the treatment of ICI myocarditis. Doms et al. reported that intravenous infusion of tocilizumab may be an efficient therapy for refractory ICI-associated myocarditis induced by a combination of nivolumab and ipilimumab [62]. Further research is needed to establish the safety and usefulness of specific immunosuppressive drugs in the treatment of ICI-associated myocarditis.

### 2.7. Anti-Microtubule Agents

Anti-microtubule agents (taxanes) are used to treat many solid tumors [4]. Taxane-associated cardiotoxicity is mainly caused by direct cardiomyocytes damage [63]. One of the most common symptoms of taxane-related cardiotoxicity is cardiac arrhythmia, which typically appears in the first few weeks of chemotherapy. Major pathomechanisms involve damage to the Purkinje system, dysregulation of autonomic nervous system control, and increased release of histamine [63]. The most common clinical presentation is bradycardia, which occurs in one-third of the patients treated with paclitaxel [64]. In rare cases, taxane therapy may induce atrioventricular block, left bundle branch block, and ventricular tachycardia due to QTc prolongation [63]. Notably, the use of antihistaminic drugs in combination with paclitaxel therapy reduced the incidence of bradyarrhythmias [65]. In addition, myocardial ischemia has been reported in patients treated with paclitaxel [63]. It should be emphasized that the concomitant use of other drugs and pre-existing CV diseases predisposes this group of patients to ischemia. It is noteworthy that polyoxyethylated castor oil used [4] as a carrier for paclitaxel in an injectable formulation may contribute to overall cardiotoxicity by inducing histamine release with subsequent coronary vasospasm and disturbance of the oxygen supply/demand balance in the coronary circulation [63].

Although taxanes are considered safer than anthracyclines, some reports indicate that docetaxel and paclitaxel can cause HF in 2.3–8% and 1–5% of patients, respectively [19]. The risk of taxane-related cardiotoxicity increases significantly in patients receiving combination therapy, especially with anthracyclines [4]. Paclitaxel may prolong the metabolism of doxorubicin and intensify its cardiotoxic effects [63]. Additionally, taxanes can cause hypertension, although much less frequently than other chemotherapeutic agents. Hypertension was found in 0.8% of patients treated with paclitaxel and in 4% of those treated with cabazitaxel [37]. In rare cases, taxanes can also cause chest pain and acute coronary events (0.2–4% of paclitaxel-treated patients) due to vasoconstriction [40]. Other anti-microtubule agents such as vinca alkaloids, especially in combination with cisplatin and bleomycin, have been found to cause chest pain in 40% of patients. The symptoms most often present in the clinical form of Prinzmetal’s angina provoked by vasospasm [40]. The main mechanisms responsible for the hyperresponsiveness of the coronary arteries are endothelial damage and dysfunction [40]. Acute coronary syndrome (ACS) due to coronary thrombosis has been reported in some patients treated with vinca alkaloids. The increased risk of coronary artery thrombosis is due to erosion, dysfunction, and damage to the endothelium and to the predominance of pro-thrombotic factors [40].

### 2.8. Vascular Endothelial Growth Factor Inhibitors

Chemotherapeutics targeting the VEGF signaling pathway are mainly used in the treatment of breast cancer, renal cell carcinoma, lung cancer, and colorectal cancer [16]. VEGF inhibitors comprise two groups: monoclonal antibodies that target the VEGF or its receptors and small molecules that inhibit the tyrosine kinases stimulated by the VEGF. It has been observed that VEGF inhibitors are associated with several CV complications. The most commonly observed complication is arterial hypertension, which may be newly induced or caused by the exacerbation of previously well-controlled blood pressure by the VEGF inhibitors [66,67]. Other adverse cardiac effects include LV dysfunction and HF (especially when VEGF inhibitors are used with conventional chemotherapies) and coronary artery thrombosis [4].

### 2.9. Tyrosine Kinase Inhibitors

TKIs are mostly used in the treatment of renal cell, breast, lung, and gastrointestinal stromal tumors (GISTs) and some leukemias (chronic myeloid leukemia and Philadelphia-positive acute lymphoblastic leukemia) [4,68,69].

TKI-induced cardiotoxicity occurs both in patients treated with VEGF inhibitors (sunitinib, sorafenib, and lenvatinib) and in some second- and third-generation BCR-ABL TKIs (ponatinib, nilotinib, dasatinib, and bosutinib) [70]. The mechanism of cardiotoxicity associated with TKIs is complex and related primarily to decreased capillary density, endothelial damage, and energy depletion [11]. Arterial hypertension, HF, arrhythmias, acute coronary syndromes, pulmonary hypertension, and venous thrombosis have been reported among patients treated with TKIs [11]. Arterial hypertension is one of the most common manifestations of TKI-associated cardiotoxicity, affecting the VEGF pathway in particular, and it usually occurs within the first few treatment cycles. The main cause for drug-induced hypertension is increased levels of endothelin-1, although activation of the renin–angiotensin–aldosterone system has also been reported as a cause [37]. The prevalence of arterial hypertension depends on the type of chemotherapy and is estimated to range from 4% (for imatinib) to 68% (for lenvatinib) [37]. The risk of CV complications is significantly higher in patients with stage II hypertension, type 2 diabetes mellitus, multiple CV risk factors, and documented pre-existing CV disease. In addition, the risk of HF is over 2.5 times higher among patients receiving VEGF-receptor TKIs [4]. LV dysfunction usually occurs in the first weeks of treatment and is much less frequently observed after a few months of treatment. It is therefore recommended to monitor patients treated with TKIs during this initial period. The incidence of HF in patients treated with sunitinib and lapatinib is estimated to be 1.5–15% and 2%, respectively [11].

An increased risk of arterial thrombosis (2.3–6%) was observed in some patients treated with anti-VEGF TKIs (especially sorafenib, sunitinib, and pazopanib) [16]. Furthermore, the majority of TKIs prolong the QTc interval; the proportion of patients with a QTc interval over 500 ms ranges between 0.2% (for bosutinib) and 8% (for vandetanib) (4). One of the most serious complications of TKI treatment is pre-capillary pulmonary hypertension. While severe, it is mostly reversible. It affects 11% of patients treated with dasatinib and typically occurs 8–40 months after treatment [4,40]. The mechanism of dasatinib-induced pulmonary hypertension is not completely understood, but it may be due to the patient’s immunological background [40].

## 3. Prevention of Chemotherapy-Induced Cardiotoxicity

The risk of cardiotoxicity should be estimated before the initiation of chemotherapy. The risk depends on the type of therapy used, the presence of CV risk factors, and the individual predisposition of the patient undergoing the treatment. Thus far, many features and factors that increase the risk of cardiotoxicity have been identified [4,20]. Some of them are specific to a particular type of anti-cancer drug (e.g., anthracyclines), although a large proportion can be extrapolated to the entire population of patients undergoing chemotherapy [4]. According to a recently published expert position statement, conducting a baseline CV risk assessment is critical for all patients referred for potentially cardiotoxic cancer therapies, so as to predict the cardiotoxicity risk, to establish a monitoring plan, and to prevent CV complications. The most important cardiotoxicity risk factors include pre-existing CV diseases, an initial elevation of cardiac biomarkers, the presence of numerous CV risk factors, and a history of treatment with cardiotoxic anti-cancer drugs or chest irradiation [70].

First, increased attention should be paid to the presence of CV risk factors and pre-existing CV diseases, with particular emphasis on hypertension, coronary artery disease, asymptomatic LV dysfunction, HF, valvular heart disease, cardiomyopathy, and arrhythmia [4,71]. Among them, the strongest risk factors for developing HF and its deterioration are age and pre-existing LV dysfunction [72]. Second, the anti-cancer drug treatment schedule, including the types of drugs, their dosages, and their formulations, should be carefully planned. Additionally, previously used therapies, especially those with anthracyclines and irradiation, must be considered [4]. Cardiotoxicity is detected through several methods, including echocardiography, nuclear cardiac imaging, cardiac magnetic resonance (CMR) imaging, and cardiac biomarkers. It is reasonable to use an individually selected chemotherapy regimen to prevent cardiotoxicity. Such regimens are highly effective in the treatment of the underlying disease and result in the least adverse impacts on the CV system [4].

Recently, Russo et al. observed that reduction in oxidative stress and amelioration of mitochondrial function as a result of phenylalanine-butyramide activity protects against experimental doxorubicin cardiotoxicity [73]. Li et al. reported that a blockade of PI3Kγ may simultaneously prevent anthracycline-induced cardiotoxicity and reduce tumor growth [74]. The adverse effects of radiotherapy involving the mediastinum should be minimized. Patients that develop HF or asymptomatic LV systolic dysfunction are likely to benefit from treatment with angiotensin-converting enzyme inhibitors (ACEIs), angiotensin II receptor blockers, and β-blockers. Thus far, the pharmacological effectiveness of primary prevention has not been confirmed in all patients undergoing chemotherapy [4,75]. The primary prevention of anthracycline-induced cardiotoxicity is controversial. The CECCY (Carvedilol Effect in Preventing Chemotherapy-Induced Cardiotoxicity) trial showed no significant changes in the LV ejection fraction during a six-month follow-up period in patients with HER-2–negative breast cancer. Carvedilol, however, was found to exert a protective effect on myocardial injury [76]. Another study (PRevention of cArdiac Dysfunction during Adjuvant breast cancer therapy or PRADA) reported no significant effect on the attenuation of the decline in the LV ejection fraction (LVEF) with the use of metoprolol in patients with breast cancer treated with anthracyclines as an adjuvant therapy. However, the trial did show significant alleviation in the LVEF decline in patients receiving candesartan [77].

Primary prevention with β-blockers and ACEIs has also been studied in patients with HER-2–positive breast cancer treated with trastuzumab as part of the MANtICORE (Multidisciplinary Approach to Novel Therapies in Cardiology Oncology Research) trial. Prophylactic use of bisoprolol and perindopril was not effective in the prevention of LV remodeling, but it did lead to attenuation in the LVEF decline in patients treated with trastuzumab [78].

## 4. Role of an Integrated Approach in the Diagnosis of Anticancer Drug-Related Cardiotoxicity

### 4.1. Imaging Modalities

Echocardiography plays an important role in the early identification and monitoring of cardiotoxicity related to anti-cancer treatment. According to a 2016 European Society of Cardiology position paper on cancer treatments and CV toxicity, baseline echocardiography should be performed in all patients undergoing potentially cardiotoxic anti-cancer therapy [4]. Two- and three-dimensional echocardiography techniques are used to assess the LVEF in specific patients prior to, during, and after anti-cancer treatment. A decrease of more than 10% in the LVEF to a value below the lower limit of normality (50–55%) suggests the presence of cancer therapy-related cardiac dysfunction, which should then be confirmed with further imaging before its categorization as either symptomatic or asymptomatic [4]. Subtle changes in the LV function can also be detected with the use of speckle tracking echocardiography (STE) and the LV global longitudinal strain (GLS) assessment [79]. This technique allows practitioners to detect impairments in LV systolic function long before the LVEF decreases. A relative percentage reduction of >15% in the LV GLS during therapy may suggest a risk of cardiotoxicity [4]. Recently published results from the SUCCOUR (Strain sUrveillance of Chemotherapy for improving Cardiovascular Outcomes) study, which included 331 anthracycline-treated patients with another HF risk factor, support the use of LV GLS in cardiac toxicity surveillance. LV GLS reduction is prognostic of a subsequent reduction in the LVEF or cancer therapy-related cardiac dysfunction, defined as a symptomatic LVEF reduction of >5% or an asymptomatic drop in LVEF of >10% compared to a baseline to <55% over a one-year follow-up period [80]. Furthermore, LV GLS-guided initiation of cardioprotective therapy results in a significantly lower reduction in LVEF compared to usual care. Keramida et al. reported that right ventricular GLS may also be used as a valuable marker of cardiotoxicity [81]. The right ventricular GLS threshold used to diagnose cardiotoxicity in that study was equal to a ≥14.8% relative reduction in GLS from the baseline value [46].

No established time intervals exist for monitoring patients for cancer therapy-related cardiac dysfunction. The current recommendations suggest repeating a cardiac evaluation every 2–3 weeks after a diagnostic echocardiography to confirm the presence of chemotherapy-related toxic effects on the myocardium [4]. Echocardiography should also be performed in follow-up appointments to monitor the patient’s systolic and diastolic function [4,82]. There are no standard recommendations, however, regarding the length of the observation period or the frequency of echocardiography evaluation in clinical scenarios.

Echocardiography also plays an important role in detecting other CV complications associated with cancer treatment, such as valvular heart disease, pulmonary hypertension, and pericardial diseases [4].

### 4.2. Cardiac Magnetic Resonance

CMR imaging is a non-invasive modality that offers cardiac images with excellent quality and reproducibility. It enables a precise assessment of cardiac function in patients being treated for cancer [83]. Typically, it is used when other imaging modalities provide inconclusive results. CMR imaging is considered a gold-standard method for evaluating the chamber size, right ventricle morphology and function, wall motion abnormalities, systolic and diastolic function, valvular function, cardiac masses, and the pericardium. It is also used to detect scarring or fibrosis [4,54]. The assessment of heart function using CMR imaging is particularly important in infiltrative cardiomyopathies, wherein the cardiac dysfunction occurs due to the deposition of various pathological substances in the myocardium. The most commonly observed infiltrative cardiomyopathies in patients treated for neoplasm are amyloidosis and hemochromatosis [84]. Accumulation of amyloids in the myocardium leads to severe diastolic dysfunction, arrhythmias, and impaired coronary blood flow [52]. CMR imaging is particularly helpful in establishing a diagnosis of amyloidosis in the early stages of the disease, when no myocardial thickening can be observed or when echocardiography is inconclusive [83]. The typical features of amyloidosis in CMR scans include a global transmural or subendocardial late gadolinium enhancement pattern, increased native T1 values (pre-gadolinium contrast), and elevated extracellular volume in post-contrast T1 mapping [82,83,85]. Moreover, CMR imaging can be used in the detection of secondary hemochromatosis in patients with malignancies who receive repeated blood transfusions [82].

It has been observed that iron overload leads to cardiac, liver, and endocrine dysfunction, and CMR imaging is the preferred imaging technique to detect this because it enables a qualitative evaluation of iron deposits in the myocardium [83,86]. It has been shown that T2 relaxation time decreases as the iron deposits in the myocardium increase [87].

In addition, CMR imaging offers unique tissue characterization techniques, which can play a significant role in diagnosing inflammation and oedema in the myocardium for patients treated for cancer. Native T1 and T2 mapping can be valuable in detecting and monitoring cardiac involvement with cancer-related therapies, as it provides evidence of early inflammatory involvement, interstitial fibrosis, and remodeling [88]. The earliest cancer chemotherapy-related cardiac abnormalities can be seen with CMR imaging within weeks after treatment and include myocardial edema and diminished LVEF. It should be highlighted that in chronic settings, typically months to years after cancer therapy-related cardiotoxicity has occurred, impaired LVEF, myocardial fibrosis, and decreased LV mass index can be observed [82].

The main limitations of this imaging modality are limited availability, long acquisition times, and its relatively high cost compared to echocardiography or multigated acquisition [48].

### 4.3. Blood Biomarkers

A variety of biomarkers have recently been introduced to improve early diagnosis and to monitor cardiotoxicity. Only a few are commonly used in the management of patients undergoing chemotherapy, including troponin I and natriuretic peptides [B-type natriuretic peptide (BNP) and N-terminal pro B-type natriuretic peptide (NT-proBNP)] [4,44,89]. However, only a few studies have investigated the role of established or novel cardiac biomarkers in stratifying patients at risk [4,26]. Most authors have found a significant relationship between elevated troponin I and the development of LV dysfunction in patients undergoing chemotherapy with various anti-cancer drugs, including trastuzumab, lapatinib, sunitinib, and anthracyclines [16,50,89,90,91]. In addition, patients with increased troponin I during treatment exhibit a significantly higher risk of adverse CV events [4,16]. Notably, the value of troponins in predicting MACE and the risk of CV death in patients treated with TKIs has not been confirmed [38].

Many studies have shown the usefulness of natriuretic peptides, especially anthracyclines and TKIs, in predicting cardiotoxicity associated with anti-cancer treatment [4,38,50,66,92]. However, more research is needed to better understand the pathophysiologies of CV biomarkers in cancer patients undergoing anti-cancer treatment [52,73]. Recently, researchers have focused on the usefulness of other biomarkers in cardio-oncology. Among them are markers of oxidative stress (myeloperoxidase), wherein elevated levels correlate to an increased risk of cardiotoxicity [93]. Moreover, according to Haybar et al., changes in some microRNA expressions play important roles in arsenic trioxide-induced cardiotoxicity and may become a promising preventive strategy in the future [10]. Other authors have confirmed the predictive values of miRNAs-29b, miRNAs-499, and miRNA-1 in patients treated with anthracyclines [16,94,95]. Some novel biomarkers, including circulating _mt_DNA, telomere length, and telomerase activity, may be useful in the early detection of chemotherapy-induced cardiotoxicity [96]. Recently, Bauckneht et al. performed fluorodeoxyglucose (FDG) positron emission tomography (PET) scans of neuroblastoma in mice models and confirmed significant correlations between FDG uptake and oxidative stress indexes. Their work supports the potential of FDG-PET as an early biomarker of doxorubicin-related cardiotoxicity [97]. A limited amount of FDG-PET scan data in cancer patients supports the idea of a metabolic change associated with distinct cancer subtypes or therapies. Although many authors have investigated the roles of ST-2, galactine-3, and GDF-15, future studies are necessary to establish their clinical efficiencies [54].

## 5. Treatment of Cardiovascular Complications during Cancer Therapy

The treatment of CV complications depends on their type, their severity, and the chemotherapy used. In some cases, the symptoms of cardiotoxicity are completely or partially reversible after the discontinuation of chemotherapy. Some patients, however, require long-term treatment [4]. In any case of cardiotoxicity, consideration should be given to reducing the dosage, method, and duration of chemotherapy or switching to another drug with similar efficacy, in addition to any possible benefits and risks (4). Two of the most difficult therapeutic issues involve asymptomatic LV dysfunction (LVEF < 50%) and HF, which may occur after treatment not only with anthracyclines, but also with other drugs, such as alkylating agents (e.g., trastuzumab) [4] or VEGF receptor TKIs [98]. Patients who develop HF or asymptomatic LV systolic dysfunction are likely to benefit from treatment with ACEIs, angiotensin II receptor blockers, or β-blockers [4]. However, large-scale randomized trials are needed to confirm this strategy. Patients who develop arterial hypertension after treatment with VEGF inhibitors should receive antihypertensive treatment [99].

QT prolongation is one of the most important preventable CV complications and must be recognized in all patients undergoing anti-cancer treatment. There are several risk factors that can cause QT prolongation, namely, anti-cancer drugs, coexisting risk factors, concomitant treatment, and side-effects associated with cancer therapies (Table 2) [100]. To improve management of QT prolongation, it is important to combine preventive measures with regular QT measurement. In patients with QT prolongation, modifications to and/or discontinuation of treatment may be suggested based on the degree of QT prolongation [100].

Interventional cardiology plays an important role in cardio-oncology. In patients undergoing fluoropyrimidine (e.g., 5-fluorouracil) therapy, symptoms of ischemia can appear in the form of angina or acute coronary syndromes, which may require treatment with percutaneous coronary intervention [4,41]. In addition, optical coherence tomography and intravenous ultrasound may provide useful information regarding early discontinuation of double antiplatelet therapy in patients with cancer and a drug-eluting stent who require cancer-related surgery or other invasive procedures [101]. Data concerning the use of percutaneous interventions guided by fractional flow reserve (FFR) and instantaneous wave-free ratio (iFR) in cancer patients are scarce. It is believed, however, that the implementation of FFR and iFR can reduce the number of unnecessary coronary interventions and double antiplatelet therapy, thereby lowering the risk of bleeding complications in cancer patients [101].

Patients with a history of cancer treatment or those undergoing anti-cancer therapy who present with acute myocardial infarction (AMI) have a greater burden of comorbidities compared to those without cancer. It has been observed that most cancer patients with AMI are treated conservatively without PCI, with in-hospital mortality remaining high and with worse clinical outcomes compared to patients without cancer [17].

Special attention should be paid to patients who have experienced thromboembolic events. Cancer is an important risk factor for thromboembolic complications, which are significantly increased during treatment with some anti-cancer drugs [73]. It is important to compare the risk of thromboembolism to the risk of bleeding for each individual patient, since cancer patients also have a high risk of bleeding [4]. Thrombocytopenia, defined as a platelet count below 100,000/mm^3^, is observed in 10–25% of cancer patients [101,102,103]. According to the Society for Cardiovascular Angiography and Interventions, cancer patients with thrombocytopenia undergoing coronary angiography do not require prophylactic platelet transfusion unless recommended by their oncology team due to a platelet count below 200,000/mL and one of the following criteria: (a) a high body temperature, (b) leukocytosis, (c) a rapid decrease in the number of platelets, or (d) other coagulation impairments. A prophylactic platelet transfusion may also be recommended if the platelet count is below 20,000/mL in patients receiving treatment for solid gynecological, colorectal, necrotic, or bladder tumors or melanoma [104]. It is believed that a platelet count above 40,000–50,000/mL means that it is safe to perform the majority of coronary interventions in cancer patients [104].

Anticoagulation in cancer patients with thrombocytopenia is controversial. Therapeutic anticoagulation is recommended to treat acute thrombosis in patients with hematological malignancies and a platelet count ≥50 × 10^9^/L [105]. In cancer patients with a platelet count below 50 × 10^9^/L, the optimal anticoagulation doses are not known, and the decision to apply therapeutic treatment should be made on an individual basis [106]. For severe thrombocytopenia and acute venous thrombosis, platelet transfusions to increase the platelet count above 50 × 10^9^/L [107] or the placement of a retrievable inferior vena cava filter may be considered [108]. The decision to reduce the dose of low-molecular-weight heparin (LMWH) in cancer patients with chronic venous thromboembolism and a platelet count below 50 × 10^9^/L should be made on an individual basis, as data to support any particular clinical strategy are still lacking. In the case of sub-acute or chronic venous thromboembolism and a platelet count below 50 × 10^9^/L, a half dose or a prophylactic dose of LMWH should be considered [109].

In any case of cardiotoxicity associated with chemotherapy, oncologists and cardiologists should cooperate closely with the aim to optimize oncological treatment while using an effective, individualized therapy to tackle CV complications. A stepwise approach to CV assessment prior to, during, and after chemotherapy is proposed in Figure 2.

Finally, it is important to emphasize the role of cardio-oncological rehabilitation (CORE) and structured exercise in cancer patients. In addition to improving the circulatory and respiratory capacity in cancer patients, CORE also aims to identify early on those patients who are at risk of developing cardiac dysfunction and to treat cardiovascular complications associated with cancer therapy [110,111].

## 6. Summary

Cardio-oncology is one of the most dynamically developing interdisciplinary fields in medicine. Effective anti-cancer treatments may be associated with increased risk of complications in many organs and systems, including the heart. Early identification of patients at high risk of cardiotoxicity and the prevention of possible CV complications are crucial elements of care in patients undergoing cancer therapy. Optimization of multimodal imaging for the early detection of cardiac involvement and rapid implementation of effective treatments are needed. These developments may positively affect the treatment of cancer and improve the prognosis of cancer patients. Despite recent progress in cardio-oncology, additional research is needed to understand the different mechanisms involved.

## Figures and Tables

**Figure 1 jcm-10-01647-f001:**
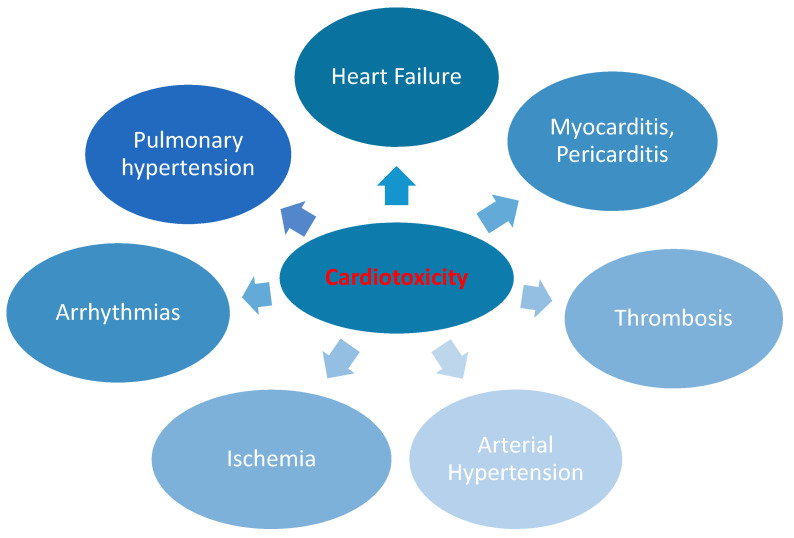
Cancer therapy-related cardiotoxicity [4,9].

**Figure 2 jcm-10-01647-f002:**
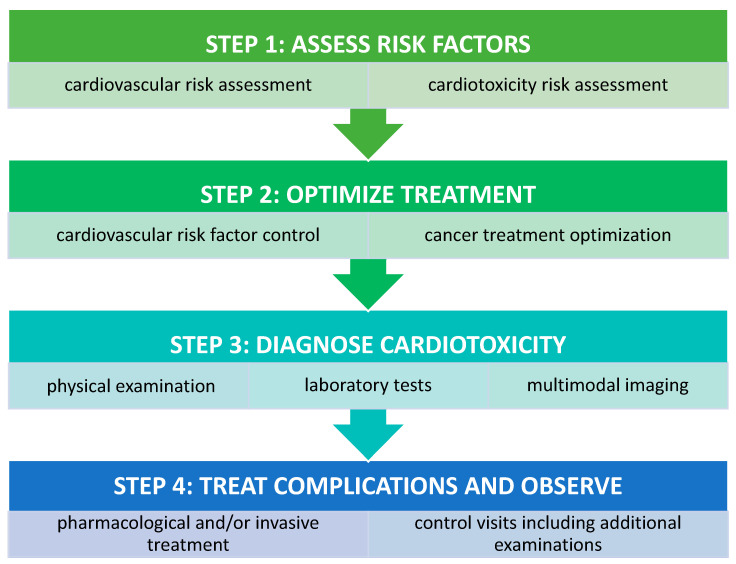
Cardiovascular evaluation in cancer patients [8,43].

**Table 1 jcm-10-01647-t001:** Major anticancer agents—mechanisms and clinical manifestations of cardiotoxicity [4,10,11,12,13,14,15,16,17,18].

Cancer Therapy	General Indications for Therapy	Main Mechanisms of Cardiotoxicity	Major Clinical Manifestations of Cardiotoxicity
Anthracyclines	Solid tumors, hematological neoplasms, pediatric cancers	Topoisomerase IIβ inhibition preventing DNA synthesis Free radical generation and oxidative stress Iron overload	HF (3–48%), AF (10.3%), Acute pericarditis (2–5%)
Alkylating agents	Solid tumors, hematological neoplasms, pediatric cancers	DNA damage Endothelial dysfunction Thrombosis	HF (0.5–28%), AF (6.6–22.5%), Acute pericarditis (2–5%)
Proteasome inhibitors	Multiple myeloma	Dysregulation of protein homeostasis in cardiomyocytes	HF (4–25%), hypertension (6.5–12%), arrhythmias (13.3%)
Fluoropyrimidines	Gastrointestinal, breast, head, neck cancers	Endothelial dysfunction Thrombosis Vasospasm Accumulation of toxic metabolites	CAD (7–10%), ACS (18%), AF (7.4–19%)
HER-2 antibodies	HER-2 overexpressing breast cancer, gastric cancer	Structural and functional changes in contractile proteins and mitochondria	HF (0.7–20%), arrhythmias (<1%)
Immune checkpoint inhibitors	Melanoma; lung, renal cell, bladder, gastric, head and neck, and Merkel cell cancers; Hodgkin’s lymphoma	Enhanced T cell activity that facilitates autoimmune reactions in the heart Hyperactivation of T cells in the heart	Myocarditis (1–2%), pericarditis (0.3%), arrhythmias (0.79%), vasculitis (0.26%)
TKIs	Philadelphia-positive leukemias; sarcomas/GIST; neuroendocrine tumors; renal cell, thyroid, and lung cancers	Endothelial dysfunction Energy depletion	HF (1–47%), AF (1.5–20%), PH (0.2–2.4%)
VEGF inhibitors	Lung, colorectal, renal cell, and esophagogastric cancers	Angiogenesis inhibition Endothelial dysfunction Arterial thrombosis	Hypertension (15.3–44.4%), HF (1.6–4%), CAD (2–8%), AF (1–10%), thromboembolism (1.4–3.8%)
Taxanes	Breast, prostate, ovarian, lung, and esophagogastric cancers	Endothelial dysfunction Arterial thrombosis	HF (1–13%), bradycardia (29%), thromboembolism (1%)

ACS—acute coronary syndromes, AF—atrial fibrillation, AMPKα—adenosine monophosphate kinase α, CAD—coronary artery disease, HER-2—human epidermal growth receptor type 2, HF—heart failure, GIST—gastrointestinal stromal tumor, PH—pulmonary hypertension, PP2A—protein phosphatase 2A, TKIs—tyrosine kinase inhibitors, VEGF—vascular endothelial growth factor.

**Table 2 jcm-10-01647-t002:** Factors increasing the risk of QT prolongation in cancer patients [100].

Risk Factor			
Anti-cancer drugs	Arsenic trioxide Ceritinib Crizotininb Dasatinib Nilotinib	Lapatinib Panobinostat Pazopanib Romidepsin Sorafenib	Sunitinib Vancetanib Vemurafenib Vorinostat
Comorbidities	Cardiac: Long QT syndrome Left ventricular dysfunction Myocardial ischemia		Non-cardiac: Hypothyroidism
Concomitant treatment	Antidepressants Antiemetics Antibiotics Antipsychotics	Anti-fungal Anti-histamines Methadone	
Cancer treatment side-effects	Nausea and vomiting Electrolyte imbalance: Hypokalemia Hypomagnesemia Hypocalcemia	Renal failure Liver dysfunction Uncontrolled diabetes	

## Data Availability

Not applicable.

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
