# Peer review of "Cancer Therapy-Related Cardiovascular Complications in Clinical Practice: Current Perspectives"

_jcm, 2021, doi:10.3390/jcm10081647_

Round 1
Reviewer 1 Report
The manuscript reads better after the extensive editing that the authors did. A few additional edits are suggested:
Abstract, lines 10-11: “Both diseases have a substantial overlap on many levels and are associated with similar metabolic alterations. Risk factors, symptoms, and outcomes are often very similar.” Since your manuscript doesn’t describe the metabolic alterations of cancer and CVD, I recommend that you remove that statement from the abstract. The sentence can be rephrased as: “Both diseases have a substantial overlap on many levels and they share common risk factors, symptoms, and outcomes.”
The medication categories reported in the table do not match the text (e.g Fluoropyrimidines are only reported in the table, proteasome inhibitors are only reported in the text, TKIs are under the same subtitle as VEGFi in the text but not in the table). Please edit accordingly so the text is consistent with the table.
Line 412: Subtitle: “Cardiovascular complications during cancer therapy”. In the paragraph you describe treatment of cardiovascular complications. Would a title like “Treatment of cardiovascular complications during cancer therapy” be more appropriate?
Please read carefully the manuscript and correct the grammatical errors.
Author Response
We would like to thank you for all suggestions which helped to improve our manuscript. We revised the manuscript as suggested. Please find below answers to all questions raised in your review:
Abstract, lines 10-11: “Both diseases have a substantial overlap on many levels and are associated with similar metabolic alterations. Risk factors, symptoms, and outcomes are often very similar.” Since your manuscript doesn’t describe the metabolic alterations of cancer and CVD, I recommend that you remove that statement from the abstract. The sentence can be rephrased as: “Both diseases have a substantial overlap on many levels and they share common risk factors, symptoms, and outcomes.”
We have modified the abstract, as suggested.
The medication categories reported in the table do not match the text (e.g Fluoropyrimidines are only reported in the table, proteasome inhibitors are only reported in the text, TKIs are under the same subtitle as VEGFi in the text but not in the table). Please edit accordingly so the text is consistent with the table.
We made changes to the text and tables. We provided a section about fluoropyrimidines (page 5).
Line 412: Subtitle: “Cardiovascular complications during cancer therapy”. In the paragraph you describe treatment of cardiovascular complications. Would a title like “Treatment of cardiovascular complications during cancer therapy” be more appropriate?
We changed the subtitle, as suggested.
Please read carefully the manuscript and correct the grammatical errors.
The manuscript has been extensively edited and proofread by the English Editing Service to ensure readability, diction, clarity and grammatical correctness.
Reviewer 2 Report
This is very interesting review article about new field of cardiology that attracts attention. The article summarized mechanism of cardiotoxicity of frequently used chemotherapeutic agent including newer agents such as immune check point inhibitor. And also, summarized important data from recent studies about preventive, diagnostic and treatment strategies for cardiovascular complications during cancer therapy.
There are some minor comments for the authors.
In paragraph "4. Role of integrated approach ~", adding subtitle "Imaging modalities" in front of first sentence "Echocardiogrphy ~" would be better, I think. Because "Blood biomarkers" is subtitled.
Also suggest adding "Treatment of" to title "5. Cardiovascular complication during~". "5. Treatment of cardiovascular complication during~" would better to show the content of the paragraph.
Author Response
We would like to thank you for all suggestions which helped to improve our manuscript. We revised the manuscript as suggested. Please find below answers to all questions raised in your review:
In paragraph "4. Role of integrated approach ~", adding subtitle "Imaging modalities" in front of first sentence "Echocardiogrphy ~" would be better, I think. Because "Blood biomarkers" is subtitled.
We added subtitle "Imaging modalities", as suggested.
Also suggest adding "Treatment of" to title "5. Cardiovascular complication during~". "5. Treatment of cardiovascular complication during~" would better to show the content of the paragraph.
We modified the title, as suggested.
Reviewer 3 Report
The paper describes the CV complications of chemotherapy, analyzing the effects of the single classes of drugs. Therefore it is of great interest for clinical practice. However some observations and integrations are indispensable for the acceptance of the paper.
- The authors on page 3 write: The effects on the heart are dose-dependent: a total dose of 400 mg / m2 of doxorubicin is known to cause cardiotoxicity . However the guidelines1 indicate a treatment with high-dose anthracycline (e.g., doxorubicin ≥250 mg / m2) as potentially cardiotoxic. It is advisable to reconcile this statement.
- Immune checkpoint inhibitor associated cardiotoxicity. For this class of drugs, in addition to the effects mentioned by the authors, a mechanism of accelerated atherosclerosis is hypothesized which increases the risk of CVD by 3 times2; probably because they can promote rapid plaque growth through plaque-mediated T lymphocyte reactivation and infiltration and increased macrophage apoptosis3.
- Anti-microtubule agents. A broader description of the pathophysiological mechanism of cardiotoxicity is required. Vinca alkaloids can induce Prinzmetal's angina, probably by vasospamus4. For paclitaxel concurrent use of other drugs and pre-existing cardiac conditions may be risk factors for induction of ischemia.The polyoxyethylated castor oil used as a vehicle for paclitaxel in the injectable formulation and may contribute to overall cardiotoxicity by inducing histamine release5.
- Antimetabolites. 5-fluorouracil (5-FU) and its oral prodrug capecitabine are not described in the review, yet can induce chest pain, ischemia, myocardial infarction, arrhythmias, and sudden death. Also in this case the vasospasmus seems to be the main mechanism5; however, microthrombotic occlusions, mediated by endothelial damage, also appear to be involved6. Other hypothesized mechanisms are: increased metabolism leading to energy depletion and ischemia, oxidative stress causing cellular damage and diminished ability of RBCs to transfer oxygen resulting in myocardial ischemia7.
- QT Prolongation. The inclusion, in the review of a separate chapter is mandatory. Emphasize the role of concomitant conditions (nausea, vomiting, and diarrhea with loss of potassium and magnesium), and therapies (antiemetics, H2-blockers, proton pump inhibitors, antimicrobial agents, and antipsychotics). Only bosutinib and vandetanib are mentioned in the paper; it is necessary to increase this list. See the paper by Coppola et al.
- On page 11 it would be useful to indicate in addition to the indications regarding the management of thrombocytopenia, also how thrombocytopenia can modulate anticoagulant therapy and low molecular weight heparin therapy in patients with cancer and thrombotic or thromboembolic phenomena.
- A brief discussion of the impact of cancer on reperfusion treatment of acute myocardial infarction should also be included on page 10. See the paper by Bharadwaj et al9.
- Finally, for the management of CV complications during cancer therapy, it is necessary to underscore the role of physical activity and structured exercise for at least some types of cancer. The Cardio-Oncology Rehabiliation, derived from Cardiac Rehabiliation, is now considered an optimal tool for treating CV complications of chemotherapy and radiotherapy and / or CVDs pre-existing or appearing after cancer diagnosis1,10
In conclusion, the work has elements of interest, however the recommended additions are mandatory
Author Response
We would like to thank you for all suggestions which helped to improve our manuscript. We revised the manuscript as suggested. All changes in the text will appear in blue. Please find below answers to all questions raised:
- The authors on page 3 write: The effects on the heart are dose-dependent: a total dose of 400 mg / m2 of doxorubicin is known to cause cardiotoxicity . However the guidelines1 indicate a treatment with high-dose anthracycline (e.g., doxorubicin ≥250 mg / m2) as potentially cardiotoxic. It is advisable to reconcile this statement.
We changed the statement, as suggested.
2. Immune checkpoint inhibitor associated cardiotoxicity. For this class of drugs, in addition to the effects mentioned by the authors, a mechanism of accelerated atherosclerosis is hypothesized which increases the risk of CVD by 3 times; probably because they can promote rapid plaque growth through plaque-mediated T lymphocyte reactivation and infiltration and increased macrophage apoptosis.
We provided additional information concerning immune checkpoint inhibitor cardiotoxicity (page 6).
3. Anti-microtubule agents. A broader description of the pathophysiological mechanism of cardiotoxicity is required. Vinca alkaloids can induce Prinzmetal's angina, probably by vasospamus. For paclitaxel concurrent use of other drugs and pre-existing cardiac conditions may be risk factors for induction of ischemia.The polyoxyethylated castor oil used as a vehicle for paclitaxel in the injectable formulation and may contribute to overall cardiotoxicity by inducing histamine release.
We provided additional information concerning anti-microtubule agents cardiotoxicity (pages 6-7).
4. Antimetabolites. 5-fluorouracil (5-FU) and its oral prodrug capecitabine are not described in the review, yet can induce chest pain, ischemia, myocardial infarction, arrhythmias, and sudden death. Also in this case the vasospasmus seems to be the main mechanism; however, microthrombotic occlusions, mediated by endothelial damage, also appear to be involved. Other hypothesized mechanisms are: increased metabolism leading to energy depletion and ischemia, oxidative stress causing cellular damage and diminished ability of RBCs to transfer oxygen resulting in myocardial ischemia.
We provided information concerning antimetabolities cardiotoxicity, as suggested (page 5).
5. QT Prolongation. The inclusion, in the review of a separate chapter is mandatory. Emphasize the role of concomitant conditions (nausea, vomiting, and diarrhea with loss of potassium and magnesium), and therapies (antiemetics, H2-blockers, proton pump inhibitors, antimicrobial agents, and antipsychotics). Only bosutinib and vandetanib are mentioned in the paper; it is necessary to increase this list. See the paper by Coppola et al.
We included a separate chapter about QT prolongation and provided Table 2 where we presented factors increasing the risk of QT prolongation (pages 11-12).
6. On page 11 it would be useful to indicate in addition to the indications regarding the management of thrombocytopenia, also how thrombocytopenia can modulate anticoagulant therapy and low molecular weight heparin therapy in patients with cancer and thrombotic or thromboembolic phenomena.
We provided additional information regarding management of thrombocytopenia and anticoagulation (page 13).
7. A brief discussion of the impact of cancer on reperfusion treatment of acute myocardial infarction should also be included on page 10. See the paper by Bharadwaj et al.
We included a discussion on reperfusion treatment in myocardial infarction (page 12)
8. Finally, for the management of CV complications during cancer therapy, it is necessary to underscore the role of physical activity and structured exercise for at least some types of cancer. The Cardio-Oncology Rehabiliation, derived from Cardiac Rehabiliation, is now considered an optimal tool for treating CV complications of chemotherapy and radiotherapy and / or CVDs pre-existing or appearing after cancer diagnosis
We included brief discussion about cardio-oncology rehabilitation (page 14).
In addition, the manuscript has been extensively edited and proofread by the English Editing Service to ensure readability, diction, clarity and grammatical correctness.
Round 2
Reviewer 3 Report
Now the review is complete and well structured and ready to be published
This manuscript is a resubmission of an earlier submission. The following is a list of the peer review reports and author responses from that submission.
Round 1
Reviewer 1 Report
Bohdan et a present a review of cancer-therapy related cardiovascular complications in clinical practice with a focus on current perspectives.
- Table 1- Consider adding ranges of incidence of the various cardiotoxicities listed in this table so that readers can easily reference the main toxicities of each class of medications
- Table 1- For HER2 therapy the tumor listed is advanced metastatic breast cancer. HER2 inhibition is used in HER2 overexpressing tumors and this should be changed.
- Anthracycline section- some discussion of the use of angiotensin converting enzyme inhibitors, betablockers should be given. There are trials showing potential benefit of these agents and some that have shown no difference. Consider including CECCY, MANTICORE, PRADA trial etc….
- HER2 inhibition section- It should be highlighted that the highest rate of toxicity reported by Slamon et al for combination anthracycline and HER2 inhibition occurred when they were given concomitantly and that is why now anthracyclines and HER2 inhibitors are given as sequential therapy.
- Immune checkpoint inhibitors- more discussion on current therapies for ICI myocarditis should be mentioned such as the use of alemtuzumab, abatacept, tocilizumab. These are the more novel and current practices being considered.
- VEGF inhibitors- this section lacks any distinction between antibody versus small molecule TKI VEGF inhibitors and the differences encountered. Also more is needed to discuss the risk of venous and arterial thrombosis in these patients.
- In the treatment section there is no discussion on interventional cardiology management and documents such as the SCAI consensus on antiplatelet management on thrombocytopenia.
Reviewer 2 Report
This review article by Bohdan et al., summarizes current knowledge on cardiotoxicity associated with commonly used chemotherapeutics.
The topic is timely and clinically relevant.
Major points
- For anthracyclines you talked about the timing of cardiotoxicity development. Can you do the same for the rest of the groups of drugs?
- Table 1 lists tyrosine kinase inhibitors, taxanes and proteasome inhibitors. Those groups of drugs are not discussed in the text of the manuscript. Could you add one paragraph on each?
- It would help if you presented available evidence on the treatment of cardiovascular complications specific to each cancer treatment group e.g. treatment of immunotherapy related cardiotoxicity (one paragraph on each, similar to what you did in the first part of the manuscript)
- The manuscript has several grammatical errors. Some, but not all, are listed below. Please read carefully and correct.
Minor points
- Line 21: Replace “discussed” with “discusses”
- Suggest to delete the word “unacceptably” in line 29
- Line 30: Add coma after “survivors”
- Line 34: Please rephrase the following sentence for clarity: “In this review current view on the cardiotoxicity associated with anti-cancer therapies with the exception of radiotherapy are presented.”
- Line 71: “and frequently occur in up to 48% of cancer patients”. Suggest to delete word “frequently”.
- Line 72: Add coma after “400mg/m2”
- Line 77: “for the cardiotoxic effects in of these drugs”
- Line 77: “reactive oxygen species (ROS) production induced by anthracyclines cause the effects on the heart”
- Line 78: “topoisomerase II beta (TOP2B) is disabled disabling, which causes causing DNA double-strand breaks and ultimately leads to mitochondrial dysfunction, activation of p53 tumor-suppressor protein, and also ROS
- Line 78: “topoisomerase II beta (TOP2B) is disabled causing DNA double-strand break that ultimately leads to mitochondrial dysfunction, activation of p53 tumor-suppressor protein, and also ROS [15]. It was shown in mice that genetic deletion of TOP2B leads to cardioprotective effects.” The way the above two sentences are phrased is a little confusing for the reader. It reads that TOP2B disabling is cardiotoxic but then genetic deletion of TOP2B is cardioprotective. Please rephrase for clarity.
- Line 80: “It was has been shown”
- Line 85: Add coma after “time of onset”
- Please rephrase the following phrase for clarity “associated with a single dose with an onset of clinical symptoms” in line 86
- Line 92: “used in conditioning in of patients”
- Line 93: “ In addition, alkylating agents are also used in the treatment of patients with multiple myeloma”
- Line 95: Please replace “Cyclophosphamide mechanism of action” with “ The mechanism of action of cyclophosphamide..”
- Line 97: “cardiomyocytes apoptosis”
- Table 1: HER-2 antibodies. Can you add in the table that they are used for gastric cancer as well?
- Line 129: “The clinical phenotypes include arrhythmias, acute heart failure and hypotension, associated to the occurrence in the setting of vasculitis, peri- and myocarditis. whereas Myocarditis is regarded as the most critical complication”.
- Line 131: “The occurrence of myocarditis depends on the used checkpoint inhibitor used and reaches up to 1.3% in combination therapy”
- Line 134: “may approximately reach” Suggest to delete “approximately”
- Line 152: “Prevention is better than treatment: how can we assess and optimize potential risk factors for of chemotherapy-induced cardiotoxicity”
- Line 166: “carefully analysed planned”
- Line 203: Add coma after “cardiac dysfunction”
- Line 203: “The topic is timely and clinically relevant. it is currently recommended to perform repeated cardiac evaluation in 2-3 weeks since the diagnostic echocardiography in order to confirm chemotherapy-related toxic effects on myocardium” Please rephrase for clarity.
- Line 241: “The earliest chemotherapy-related CV abnormalities in CMR can be seen within weeks after chemotherapy and include myocardial edema and diminished LVEF while in chronic setting typically months to years after cancer therapy-related cardiotoxicity - impaired LVEF, myocardial fibrosis, and a decreased LV mass index can be observed” Please split this long sentence into two.
- Line 281: Under the paragraph titled “Cardiovascular complications during cancer therapy” you are talking about treatment of CV complications. You may want to change the title to express that.
- Line 311: “Cardiovascular complications during cancer therapy”. You used the same title as the paragraph above. This paragraph is the same as the Summary paragraph. Please delete.